# Antigenotoxic, Anti-photogenotoxic, and Antioxidant Properties of *Polyscias filicifolia* Shoots Cultivated In Vitro

**DOI:** 10.3390/molecules25051090

**Published:** 2020-02-28

**Authors:** Ramona Figat, Anita Śliwińska, Anna Stochmal, Agata Soluch, Magdalena Sobczak, Anna Zgadzaj, Katarzyna Sykłowska-Baranek, Agnieszka Pietrosiuk

**Affiliations:** 1Department of Environmental Health Science, Medical University of Warsaw, Faculty of Pharmacy, Banacha 1, 02-097 Warsaw, Poland; rfigat@wum.edu.pl (R.F.); magda.sobczak@op.pl (M.S.); azgadzaj@wum.edu.pl (A.Z.); 2Department of Pharmaceutical Biology and Medicinal Plant Biotechnology, Medical University of Warsaw, Faculty of Pharmacy, Banacha 1, 02-097 Warsaw, Poland; katarzyna.syklowska-baranek@wum.edu.pl (K.S.-B.); agnieszka.pietrosiuk@wum.edu.pl (A.P.); 3Department of Biochemistry and Crop Quality, Institute of Soil Science and Plant Cultivation, State Research Institute, 24-100 Puławy, Poland; asf@iung.pulawy.pl (A.S.);

**Keywords:** antigenotoxic agents, chlorogenic acid, ferulic acid, caffeic acid, Salmonella typhimurium

## Abstract

Traditional medicinal plants are an important source of active compounds with potential antimutagenic activity. *Polyscias filicifolia* Bailey (Araliaceae) is a South Asian traditional herb used as an adaptogenic and cardiac drug. Extracts of *P. filicifolia* contain a wide range of biologically active compounds like phenolic acids and triterpenoid saponins. In the present study. antigenotoxic potential of three naturally occurring phenolic acids and extracts of *P. filicifolia* growing in vitro with the addition of elicitors was evaluated against direct (4-nitroquinoline-N-oxide (4NQO) and mitomycin C (MMC)) and indirect mutagens (2-aminoanthracene (2AA)). The evaluation was made using a bacterial *umu*-test. Moreover, the ability to prevent photogenotoxicity induced by chlorpromazine (CPZ) under UVA irradiation was measured. The phytochemical profiling of examined extracts revealed the presence of numerous compounds with the prevelance of chlorogenic, caffeic, and ferulic acid derivatives; however, saponin fractions were also determined. The antioxidant potential of extracts strictly correlated with their composition. The tested extracts exhibited high antigenotoxic activity if the assay was performed with 2AA and metabolic activation. Moreover, the extracts slightly decreased the MMC-induced genotoxicity. However, an increase of the genotoxic effect was observed in the assay performed with 4NQO. In addition, photo-antigenotoxic activity was observed. In our study, phenolic acids exhibited lower activity than the extracts.

## 1. Introduction

Human cells and tissues are constantly exposed to oxidative stress generated both endogenously (produced during biochemical processes) and exogenously (originating from the environment, e.g., UV light and pollution resulting from industrial development). Although reactive oxygen species (ROS) are involved in numerous physiological processes, they have to be kept under strict control by inner enzymatic antioxidant systems and free radical scavengers; otherwise, the loss of equilibrium may lead to development of serious diseases like cancers [1], cardiovascular, and neurodegradative disorders [2,3] and many other age-related conditions [4].

Among a great number of bioactive compounds present in vegetables, fruits, and herbs, phenolic compounds attract attention due to their antioxidant as well as antimutagenic properties and their preventive role in diseases associated with oxidative stress [5,6,7,8,9,10,11,12,13,14,15,16,17]. In recent years, the use of herbal medicines has grown considerably, not only for their beneficial effect on human health but also due to their activity as antiaging agents [18,19]. The constant growth of market demand for products of plant origin is observed. Plant products are considered healthy due to their origin, recognized as natural and/or organic and derived from cultivation applying eco-friendly farming practices. Plant biotechnology can help cover the increasing demand for plant-derived products. Plant in vitro cultures can serve as a reasonable alternative to harvest from natural environments with no limitation due to the slow growth, with cultivation under controlled conditions, and with independence from seasonal changes. Furthermore, the multiplication of selected, highly productive cells or organs directly engaged in biosynthesis of desired compounds can allow the lowering of their downstream extraction and purification costs [20,21]. One of the most rapidly developing research areas in plant biotechnology is plant stem cell cultures for cosmetic formulation (See Reference [22] and references herein). The application of stem cells derived from a cultured plant that is callus and/or cell suspension cultures with fresh, freeze-dried cells; extracts; or isolated compounds can reduce wrinkle depth, enhance elastin synthesis, and reduce sebum production rate [22]. However, the main concern is stability and reproducibility in the composition of stem cells or their extracts resulting from in vitro cultures. Not only the phytochemical profile but also the antigenotoxic, anti-photogenotoxic, and antioxidant activity of extracts should be verified to ensure safety of their usage. To increase the concentration of desired constituents in cells and tissues cultivated in vitro, the most frequent elicitation strategies are employed [23,24,25] enhancing synthesis and accumulation of desired secondary metabolites. Phenolic compounds represent the desired group of chemical constituents of plant stem cells used in various cosmetic formulation due to their properties connected with promoting skin photo protection, enhancing skin cell renewal, stimulating collagen and elastin synthesis, and many others [19,26].

*Polyscias filicifolia* Bailey (Araliaceae) is a traditional medicinal herb of Southeast Asia. This species is known as fern leaf Panax and has been included into Vietnamese Pharmacopeia, and extracts of the plant are used as a tonic, adaptogenic, and antibacterial agent and contain a wide range of biologically active compounds like phenolic acids, flavonoids, and saponins (Śliwińska et al. and publications herein [27]). The chemical composition of *P. filicifolia* has not been fully recognized yet. Recently, the efficient biotechnological method for enhanced phenolic compound production in shoots of *P. filicifolia* cultivated in vitro has been developed [27]. The most abundant in extracts prove to be chlorogenic acid (CGA), and elicitation with salicylic acid (SA) or methyl jasmonate (MeJA) caused a 2- or 3-fold increase in its accumulation up to 5.03 mg/g dry weight, which was remarkably higher than in other plants recognized as rich sources of CGA. Preliminary investigations of biological activity of *P. filicifolia* ethanolic extracts derived from in vitro cultivated shoots, their leaves, cell biomass, and saponin fraction revealed that they possessed no genotoxic properties [28].

As the elicitation could significantly enhance concentration or even change the profile of phytoconstituents in plant tissues and resulting extracts, simultaneously influencing their biological properties, the major goal of the present study was the assessment of the antigenotoxic, anti-photogenotoxic, and antioxidant potential of *P. filicifolia* methanolic extracts (EXTs) derived from shoots subjected to elicitation. The following *P. filicifolia* EXTs were tested: control extract (EXT0) and extracts of plants treated with different elicitors, such as salicylic acid (EXT(SA)) and methyl jasmonate (EXT(MeJA)), and with both elicitors (EXT(MeJA + SA)). Moreover, the antigenotoxic and anti-photogenotoxic properties of three phenolic acids (PhAs), CGA, caffeic acid (CA), and ferulic acid (FA), was estimated and compared with plant extracts. The three known mutagens—4NQO, MMC, and 2AA—and one known photogenotoxin (CPZ) were used in the bacterial *umu*-test. In addition, two cytotoxicity assays were performed to estimate the possible cytotoxic effect of EXTs towards the V79 cells. The chemical profiling and antioxidant potential of plant extracts was also performed.

## 2. Results

The four extracts prepared from shoots of *P. filicifolia* growing in vitro in the presence of selected elicitors or in untreated cultures (control) were subjected to phytochemical profiling and to examination of their antioxidant and antigenotoxic potential. The latter was compared with antigenotoxic activity of standard compounds: CGA, FA, and CA. Moreover, cytotoxicity of EXTs was evaluated towards V79 cells. The content of CGA, CA, and FA is presented in Table 1. Phytochemical analysis of extracts showed significant changes in the production of CGA and the total amount of polyphenols and flavonoids (Table 1 and Figure 1). The most abundant phenolic acid was CGA, with its highest amount—4.2 ± 0.17 mg/g dry weight (DW)—resulting from elicitation with MeJA + SA, and was 3.5-fold higher than in EXT0 (1.18 ± 0.07 mg/g DW) (Appendix A). Also in this type of EXT, the highest total phenolic and flavonoid content was noted, 7.8 ± 0.21 mg GAE/g DW and 4.3 ± 0.27 mg QE/g DW, respectively. The free radical scavenging capacity was strongly correlated (the Pearson correlation test was used) with concentration of phenolic and flavonoid compounds (Figure 1, Table 1). The HPLC-DAD-UV-Vis analysis reveal the presence of large groups of chlorogenic acids consisting of esters: caffeoylquinic acids; feruloylquinic acids; caffeic, ferulic, and quinic acid derivatives; flavonoid quercetin derivatives; and procyanidins, catechins, and saponins of dammarane and oleanane types (Figure 2, Appendix A). The combined fractions of phenolics and saponins possessed the highest percentage share in EXT0, EXT(SA), and EXT(MeJA) extracts (Figure 2).

All of the tested extracts exhibit low cytotoxicity against the V79 cell line. IC${}_{50}$ values are >1 mg/mL. The highest inhibition of the viability was observed in the MTT assay for EXT0 and EXT(SA) in the concentration of 1 mg/mL: 14.3% and 13.5%, respectively (Figure 3).

In the preliminary *umu*-test, the genotoxicity of PhAs and EXTs was evaluated (Table 2 and Table 3). None of the tested concentrations exhibited the genotoxic effect (IR < 1.5).

The results of the *umu*-test performed without metabolic activation against 4NQO and MMC genotoxicity are presented in Table 4. The protective effect against 4NQO (0.25 μg/mL) was observed in treatments with EXT0 and EXT(SA) in the concentration of 0.25 mg/mL and with CA (0.3125 μg/mL). EXT(MeJA) in all of the tested concentrations increased the genotoxic effect of 4NQO, as did EXT(SA) in the concentration of 1 mg/mL and EXT(MeJA + SA) in the concentrations of 1 and 2 mg/mL. In the mixtures with 4NQO in the concentration of 0.05 μg/mL, EXTs increased the genotoxicity; however, PhAs have no significant activity.

In the test with MMC 0.02 μg/mL, the decrease of genotoxicity was observed only in the mixture with EXT(SA) (0.5 and 1 mg/mL) and EXT(MeJA) (2 mg/mL), whereas in the mixture with EXT(MeJA) (0.25 mg/mL) and FA (0.3125 μg/mL), an increase of genotoxicity was observed. In the test with MMC in the concentration of 0.004 μg/mL the reduction of genotoxicity was observed for the lowest tested concentration of EXT0.

The results on the effect of EXTs against 2AA genotoxicity are presented in Table 5. All of the tested extracts reduced the genotoxic effect of 2AA. The highest antigenotoxic potential was observed for EXT(MeJA + SA) in the concentration of 2 mg/mL (63–66%).

The results of the tested PhAs against 2AA genotoxicity are presented in Table 6. CGA in concentrations of 5 and 10 μg/mL increase the genotoxic effect of 2AA. A reduction of the genotoxic potential was observed solely for one concentration of FA (0.625 μg/mL) and one concentration of CA (1.25 μg/mL).

The anti-photogenotoxic evaluation was performed with EXT(MeJA + SA) and PhAs (Table 7). The photoprotective effect was observed: EXT(MeJA + SA) reduced CPZ-induced photogenotoxicity with inhibition of 37%. FA had a weak anti-photogenotoxic potential (11% to 20%), whereas CGA and CA were active only in the concentration of 2.5 μg/mL.

## 3. Discussion

The commercialization of plant-derived products for medicinal as well as cosmetic purposes involves studying their overall safety, including cytotoxic, antimutagenic, and antigenotoxic properties. In the present study, the extracts exhibit low cytotoxicity against the V79 cell line. The reduction of viability to 14.3% in the MTT assay was observed. [28] evaluated the cytotoxic effect of *P. filicifolia* dry shoot extracts—but not as elicited in the present research—using the murine connective tissue cell line C3H/AN—L929 on the EZ4U test. The IC${}_{50}$ value was calculated as 23.85 μg/mL. At the concentration of 62.50 μg/mL, there were no living cells in the culture. In present experiments, much higher concentrations have no cytotoxic effect to the V79 cell line (IC${}_{50}$ > 1 mg/mL). These results are consistent with the previous observations that extracts of *P. filicifolia* prepared from shoots growing in the presence of MeJA or SA were not toxic to HaCaT cells but also lightly enhanced cell viability [27].

According to results, the tested concentrations of EXTs and PhAs exhibited no genotoxic effect in the *umu*-test. In the mentioned paper [28], the Ames test and the micronucleus assay were used to evaluate the genotoxicity of *P. filicifolia* extracts and no genotoxic effect was observed. Stich et al. [29] described clastogenic activity of CGA; however, Alarcón-Herrera et al. [30] reported that CGA in the dose of 100 mg/kg in the micronuclues assay in vivo has no genotoxic effect. Maistro et al. [31] reported the genotoxicity of CA and FA in concentrations of 500 and 1500 μM. They performed the micronucleus assay and observed increases in the frequency of micronucleated cells in hepatoma tissue culture (HTC) cells. In the present study, the highest tested concentration was approximately 25-fold lower. Based on the present study, it can be concluded that the tested concentrations of extracts and PhAs are safe for DNA and exhibit no genotoxic potential.

In the present study, a significant decrease of IR was observed in antigenotoxic treatments with extracts against 2AA and MMC. EXT(MeJA + SA) exhibited the highest antigenotoxic potential against 2AA (up to 66%). On the other hand, an increase of the genotoxicity was observed in the treatment with extracts against 4NQO and CGA against 2AA. Moreover, the results suggest that the tested PhAs are probably not active against 4NQO and MMC-induced genotoxicity in tested concentrations. Based on the results, it also can be concluded that tested PhAs alone are not responsible for the antigenotoxic potential of EXTs against 2AA.

Almost all tested EXTs mostly increased the 4NQO-induced genotoxicity, with the exception of EXT0 and EXT(SA), which caused a significant reduction of genotoxicity but solely in the concentration of 0.25 mg/mL. This can be attributed to the considerable content of saponin fraction determined in EXT0 and EXT(SA) in comparison to other EXTs (Figure 2; Appendix A). This suggestion can be supported by the results of antigenotoxic saponin properties research led by Lee et al. [32], Berhow et al. [14], and Geetha et al. [33]. The mechanism of mutagenesis by 4NQO consists of its transformation by the cellular enzymes to an active compound which can covalently bind to guanine or adenine of DNA, leading to basepair substitution [34] and intracellular oxidative stress [35]. Although EXTs revealed an antioxidative potential, the increase in the 4NQO-induced genotoxicity was observed. Previously, it was reported that antioxidants did not exhibit a protective effect against 4NQO [36] or an increase in the genotoxicity [37]. This could be explained by the fact that the oxidative stress is not the only possible mechanism of action of this mutagen. The tested EXTs contain a lot of active substances, some of which may generate an additional genotoxic effect with the combination with 4NQO. It was reported that polyphenols, the most abundant fraction in the EXTs, can exert a prooxidant effect and that the possible mechanism of such action involves interaction with transition metal ions [38,39,40,41,42,43,44].

In current investigations, PhAs have no significant effect against 4NQO genotoxicity, except the lowest tested concentration of CA, which reduced the IR value of 25%. Yamada and Tomita [45] performed an Ames-test with the *S. typhimurium* TA98 strain to evaluate antigenotoxic potential of PhAs against 4NQO. All of the PhAs reduced the number of revertants: CA by 28%, CGA by 15%, and FA by 17%. Abraham et al. [46] attempted to evaluate the antigenotoxic potential of CGA using micronucleus assay in vitro. CGA in concentration of 200 μM (70.8 μg/mL) reduced the number of micronuclei induced by 4NQO (0.5 μg/mL) of 50–70%. In the present experiment, the approximately 7-fold lower concentration of CGA was applied and no antigenotoxic effect was observed. Based on these results, it could be concluded that the tested concentration of PhAs have no significant antigentoxic potential against 4NQO.

Obtained results suggested that EXTs have slight antigenotoxic potential against MMC. The inhibition of MMC-induced genotoxicity was observed for two tested concentrations of EXT(SA) and one of EXT0 and EXT(MeJA). However, no significant activity was observed for EXT(MeJA + SA), and EXT(MeJA) increased the genotoxicity in the lowest tested concentration. In the present study, PhAs exhibit no antigenotoxic activity against MMC in the tested concentrations. However, Abraham et al. [47] reported the antigenotoxic potential of CGA against MMC genotoxicity. In the performed micronucleus assay, the inhibition of 65% was observed after the addition of CGA in concentrations of 5 and 50 μM (1.77 μg/mL/17.7 μg/mL) to MMC 0.9 μM (0.3 μg/mL). Though similar concentrations were used, different results were obtained. The main biological mode of action of MMC is its ability to bind covalently to DNA [48]. Other mechanisms are alkylation of DNA [49] and free radical formation [50]. CGA may inhibit the mechanism which leads to formation of micronuclei, although it is not able to prevent DNA damage which activates the SOS-system in *S. typhimurium*.

2AA is a pre-carcinogen; its reactive electrophilic metabolites bind with DNA [51]. All of the EXTs exhibited high antigenotoxic potential against 2AA-induced genotoxicity. The antigenotoxic mechanism of EXTs may be competition with 2AA molecules for S9 fraction enzymes, binding to DNA and protection against electrophilic metabolites or interaction with metabolites. On the other hand, PhAs are probably not responsible for EXTs activity against 2AA. The quantity of CGA was evaluated as the highest; however, CGA significantly increased genotoxicity when added to 2AA.

The anti-photogenotoxic evaluation was performed for PhAs and the extract which exhibits the strongest antigenotoxic activity against 2AA (EXT(MeJA + SA)). The extract demonstrated the best inhibitory effect, which was over 34% for the highest tested concentration. FA exhibited weakened anti-photogenotoxic effect by up to 19.5%. CGA and CA were active only if the 2.5 μg/mL concentration was used. The extract and PhAs under UVA irradiation may act as a UV filter, reducing the CPZ transformation to the unstable promazyl radical. There are reports in the literature that CA and FA have partial UVA absorption properties [52,53]. Another mechanism of anti-photogenotoxic effect of the tested compounds may be free radical scavenging.

The UHPLC-DAD-MS/MS analysis of EXTs subjected to current experiments revealed the presence of numerous compounds belonging to various chemical groups like polyphenols and saponins, although PhAs were in the form of their isomers. This discrepancy between HPLC-UV-Vis and UHPLC-DAD-MS/MS analyses could be a result of their chemical instability and easy transformation as esterification and reaction with water [19]. The activity of PhAs tested in the present study was lower in comparison to the activity level of examined EXTs. This phenomenon could be attributed to the combined action of all EXTs components. Xu et al. [54] reported that CGA and its isomers exhibited antioxidant properties and effects protective against DNA damage. However, dicaffeoylquinic derivatives rather than caffeoylquinic ones proved to be the most potent depending on their structure. Nonetheless, three caffeoylquinic acid isomers, among them CGA, demonstrated quite similar antioxidant activities.

## 4. Materials and Methods

### 4.1. Plant Material

Shoots of *Polyscias filicifolia* cultivated in vitro as described previously were used for extract preparation [27]. Briefly, shoots cultivated for five weeks on Linsmaier and Skoog medium [55] supplemented with 2 mg/L 6-benzylaminopurine (BAP) and 0.5 mg/L kinetin (KIN) were transferred onto a fresh medium containing or not (control culture) elicitors: methyl jasmonate (MeJA, 200 μM), salicylic acid (SA, 50 μM), or a combination of them (MeJA 200 μM + SA 50 μM). Elicitation was carried out for one week. After that, shoots were collected, weighed, and lyophilized. Lyophilized plant material was subjected to phytochemical and biological activity investigations.

### 4.2. Chemicals

Chlorogenic acid (CGA) (CAS no. 327-97-9), caffeic acid (CA) (CAS no. 331-39-5), ferulic acid (FA) (CAS no. 1135-24-6), quercetin (Q) (CAS no. 522-12-3), gallic acid (GA) (CAS no. 149-91-7), 4-nitroquinoline N-oxide (4NQO) (CAS no. 56-57-5), 2- aminoanthracene (2AA) (CAS no. 613-13-8), DPPH (1,1-diphenyl-2-picrylhy-drazyl) (CAS no. 84077-81-6), ABTS (2,2′-azinobis(3-ethylbenzothiazoline-6- sulfonic acid) diammonium salt) (CAS no. 30931-67-0), and Trolox (6-hydroxy-2,5,7,8-tet- ramethychroman-2-carboxylic acid) (CAS no. 53188-07-1) were purchased from Sigma-Aldrich (Poznań, Poland). 4NQO and 2AA were dissolved in DMSO. Mitomycin C was purchased from Serva (Toruń, Poland). Mitomycin C was dissolved in methanol. D-glucose 6-phosphate disodium salt hydrate (G-6-P) (CAS no. 3671-99-6), disodium salt hydrate (CAS no. 3671-99-6), and 2-nitrophenyl β-D-galactopyranoside (ONPG) (CAS no. 369-07-3), which was the β-galactosidase enzyme, were purchased from Sigma Aldrich. Nicotinamide adenine dinucleotide phosphate (NADP) (CAS no. 24292-60-2) was purchased from MP Biomedicals. Sodium lauryl sulfate (CAS no. 151-21-3) was purchased from BDH chemicals (Poland). DMSO (CAS no. 67-68-5) was purchased from Avantor Performance Materials (Poland). Methanol (CAS no. 67-56-1) was purchased from Merck (Warsaw, Poland), and methanol for plant material extraction and antioxidant activity investigations was purchased in Avantor Performance Materials (Warsaw, Poland). Petroleum ether (CAS no. 8032-32-4) was bought in Avantor Performance Materials (Poland).

### 4.3. Phytochemical Investigations

#### 4.3.1. Preparation of Extracts and HPLC-UV-Vis Determination of Phenolic Compounds

Plant extracts (EXTs) were prepared according to the method described previously by Śliwińska et al. [27]. Briefly, powdered lyophilized shoots were sonicated with 100% methanol (3 × 1 mL) for 15 min at 40 ${}^{\circ }$C. After centrifugation, the extracts were collected, evaporated to dryness, redissolved in 10 mL of methanol and extracted three times with 15 mL petroleum ether for 5 min. Next, the water phase was collected and lyophilized. The dry residue was next dissolved in 100% methanol (HPLC grade) and subjected to HPLC-DAD-UV-Vis analysis on the DIONEX HPLC system (Sunnyvale, CA, USA). The gradient elution was applied: solvent B, acetonitrile; solvent A, 0.04 M orthophosphoric acid. The gradient program was as follows: 0 min, B10%; 5 min, B45%; and 15 min, B55%. The flow rate was 1 mL/min, and C18 reversed phase column (EC 250/4.6 Nucleosil 120–127 mm; Macherey-Nagel, Germany) was used. The content of three phenolic acids (PhAs) was determined: chlorogenic acid (CGA), ferulic acid (FA), and caffeic acid (CA). All data were recorded at 327 nm. Peaks were assigned by spiking the samples with the standards and by comparing the retention times and UV spectra.

#### 4.3.2. UHPLC-DAD-MS/MS Analysis

##### Preparation of Samples to Analysis

The composition of four samples originating from shoots growing under control conditions and treated with MeJA, with SA, or with MeJA and SA simultaneously were profiled using UHRMS (ultra high resolution mass spectrometry) on a Dionex UltiMate 3000RS (Thermo Scientific, Darmstadt, Germany) system with a CAD (charged aerosol detector) interfaced with a high-resolution quadrupole time-of-flight mass spectrometer (HR/Q-TOF/MS, Impact II, Bruker Daltonik GmbH, Bremen, Germany). For this purpose, 10 mg of extract from each of the four samples was taken for qualitative analyzes and dissolved in 70% MeOH to the final concentration of 10 mg/mL. Next, the samples were sonicated for 5 min at 30 ${}^{\circ }$C (sonicator SONOREX DIGITEC DT 510 H, Bandelin, Germany) and centrifugated for 5 min at 8000 rpm (laboratory centrifuge Polygen Sigma 3-16 KL, Sigma, Germany). Finally, 150 μL of the supernatant from each sample was subjected to analysis.

##### Ultra-high-resolution Mass Spectrometry Parameters

The chromatographic separation was carried out on a CORTECS T3 column (150 × 2.1 mm 120 Å, 2.7 μm, Waters). The flow rate was set at 0.6 mL/min, and the column temperature was maintained at 55 ${}^{\circ }$C. The mobile phase was composed of solvent A (ultrapure water MilliQ containing 0.1% formic acid) and solvent B (LC-MS grade acetonitrile acidified 0.1% formic acid). The chromatographic method consisted of the following linear gradient: 5% B from 0 to 0.5 min. The concentration of B was then increased to 80% from 0.5 to 27 min. Compounds were analyzed based on data from UV absorption at a wavelength range set at 195–600 nm, and mass spectra with a mass scan range was set at 50–1800 m/z. Electrospray ionization (ESI) was performed in positive and negative ion modes. The ion source parameters in negative ion mode were capillary voltage set at 3.0 kV, nebulizer at 0.7 bar, dry gas at 6.0 L/min, and dry temperature at 200 ${}^{\circ }$C. Data acquisition and processing were performed using DataAnalysis 4.3 (Bruker Daltonik GmbH, Bremen, Germany). All analytes, including phenolic compounds, were identified based on previously literature data and based on data from the library Compound Crawler Bruker Program, DataAnalysis 4.3 (Bruker Daltonik GmbH, Bremen, Germany).

#### 4.3.3. Determination of Total Phenolic Compounds Content

The Folin–Ciocalteu method was applied for determination of total phenolic compound content according to modified protocol described by Pękal et Pyrzyńska [56]. Results are expressed as gallic acid equivalents (GAE): mg GAE/g DW (dry weight). All analyses were performed in triplicate on 96-well plates using BioTek reader (EPOCH, CA, USA) at 765 nm.

#### 4.3.4. Determination of Total Flavonoid Content

Total flavonoid content was estimated according to the method described by Pękal et Pyrzyńska [56] with some modifications. The total flavonoid content is expressed as equivalents of quercetin (QE): mg QE/g DW. All analyses were performed in triplicate on 96-well plates using BioTek reader (EPOCH, CA, USA) at 425 nm.

#### 4.3.5. Determination of Antioxidant Activity

Antioxidant activity of plant extracts was determined using DPPH and ABTS assays according to the procedure described by Mareček et al. [57]. All results are calculated as Trolox equivalent (TE): mg TE/g DW. In DPPH free radical scavenging and cation-radical ABTS•+ assays, the absorbance was recorded at 517 nm and 734 nm, respectively, after 30 min of incubation. All analyses were performed in triplicate on 96-well plates using BioTek reader (EPOCH, CA, USA).

### 4.4. Antigenotoxicity Assessment

#### 4.4.1. Metabolic Activation

Metabolic activation was obtained with S9 fraction prepared from livers of male Sprague Dawley rats. Rats were treated with Aroclor 1254 (single dose of 500 mg/kg body weight) in soya oil five days before the isolation. After isolation, the evaluation of cytochrome p450 level was performed and samples were stored at −80 ${}^{\circ }$C. The S9 mix was prepared according to the method described by Maron and Ames [58].

#### 4.4.2. Bacterial Strain

*Salmonella typhimurium* strain TA1535/pSK1002 was purchased from Deutsche Sammlung von Mikroorganismen und Zellkulturen GmbH (DSMZ, Germany).

#### 4.4.3. *Umu*-Test

The *umu*-test detects the induction of the SOS system in the strain *S. typhimurium* TA1535/pSK1002. The SOS system is the bacterial response to the DNA-damaging agents. One of the genes involved in the SOS system is *umuC*. The strain used in the test carries the fusion of *umuC* and *lacZ* genes, which are placed at the plasmid pSK1002. Therefore, β-galactosidase activity strictly depends on SOS system induction level and the genotoxic activity of the tested compound [59,60]. In the present study, the *umu*-test was carried out in the micro-plate variant according to the ISOguideline [61]. β-galactosidase activity of the tested compound was presented as induction ratio (IR). IR was calculated as the β-galactosidase activity of the tested compound relative to the negative control.

IR was calculated on the basis of spectrophotometric measurement. The absorption at 420 nm indicated the intensity of the enzymatic reaction, and the optical density at 600 nm indicated the bacteria growth. The measurements were performed with an Asys UVM340 Hightech microplate spectrophotometer. The 1.5 fold and greater increase of the β-galactosidase activity resulting in IR value of 1.5 and greater indicated the genotoxicity of the sample.

#### 4.4.4. Determination of Antigenotoxicity and Anti-photogenotoxicity by the *umu*-test

The antigenotoxic potential was evaluated against the genotoxic action of three known genotoxic agents: 4NQO, MMC in the absence of metabolic activation with rat liver S9 fraction, and 2AA in the presence of metabolic activation. Firstly, the genotoxic potentials of 4NQO (0.05 and 0.25 μg/mL), MMC (0.004 and 0.02 μg/mL), and 2AA (1 and 10 μg/mL) were measured by the *umu*-test. At the same time, the genotoxicity of PhAs (10 μg/mL) and EXTs (0.25–2 mg/mL) was investigated. Next, the inhibition of 4-NQO, MMC, and 2AA genotoxicity by PhAs and Exts was investigated. PhAs were added to the genotoxins at concentrations of 0.3125–10 μg/mL. EXTs were added to the mutagens at concentrations of 0.25–2 mg/mL. The rate of antigenotoxicity (%) of the tested sample was calculated as the inhibition of IR induced by the genotoxic agent (4NQO, MMC, or 2AA) at the particular concentration according to the following equation:(1)$$Antigenotoxicity\%=(1-\frac{IR_{genotoxin+sample}}{IR_{genotoxin}})100\%$$

The anti-photogenotoxicity was determined using chlorpromazine (CPZ) as an agent inducing genotoxic response under UVA irradiation by forming an unstable promazyl radical which is able to bind to DNA [62]. The protocol developed by Skrzypczak et. al [63] was applied. CPZ was tested at concentrations of 5 mg/L and 10 mg/L by the modified *umu*-test after UVA irradiation was carried out with the lamp emitting UVA 365 nm (0.231 mW/cm${}^{2}$ ). The anti-photogenotoxic activity of CA (1.25–10 mg/L), CGA (1.25–10 mg/L), FA (1.25–10 mg/L), and EXT(MeJA + SA) (0.125–1 mg/L) was evaluated. The EXT(MeJA + SA) was chosen due to having the highest antigenotoxic potential among all tested EXTs. The rate of anti-photogenotoxicity was calculated using the same formula as the rate of antigenotoxicity.

### 4.5. Determination of Cytotoxicity

#### 4.5.1. Cell line and Culture Conditions

The Chinese hamster lung fibroblast V79 cell line (ATCC® CCL-93™) was purchased from the American Type Culture Collection. V79 cells were cultured under standard conditions in DMEM (Gibco) supplemented with 10% fetal bovine serum and antibiotics (100 IU/mL penicillin and 0.1 mg/mL streptomycin). Cells were maintained as monolayer in tissue-culture flasks at 37 ${}^{\circ }$C in a humidified atmosphere with 5% CO${}_{2}$.

#### 4.5.2. Neutral Red Uptake (NRU) Assay

Before treatments, the cells were disaggregated with 0.25% trypsin and a total of 1×10${}^{5}$ cells per well was transferred in 96-well microplates and incubated for 24 h. After this period, the medium was removed and the cells were rinsed with PBS (Gibco). Subsequently, the cells were treated with different concentrations of the tested substances and EXTs for 24 h in the serum-free medium. Following the exposure to EXTs, the cells were washed with PBS and incubated for 3 h in the DMEM containing 50 μg/mL of the neutral red (NR). The medium was discarded, the cells were rinsed with pre-warmed PBS, and the NR desorb solution (EtOH/acetic acid) was added to all wells to extract NR from the cells. The plates were shaken for 10 min, and the absorbance of the solution in each well was measured at 540 nm in the Asys UVM340 Hightech microplate spectrophotometer.

#### 4.5.3. MTT Assay

Following the exposure, the cells were washed with PBS and were incubated with MTT dissolved in the serum-free medium for 2 h. The cells were then washed with PBS, and isopropanol was added in order to solubilize formed formazan. The plates were shaken for 10 min, and the absorbance of the solution in each well was measured at 560 nm in the Asys UVM340 Hightech microplate spectrophotometer.

### 4.6. Statistical Analysis

The statistical analyses were made on the results of at least 3 independent biological replicates performed with completely fresh bacteria cultures (all measurements were performed in triplicate or more independent experiments). All data were analyzed using the STATISTICA software package. For data with normal distribution (tested with Shapiro–Wilk test), Student’s t-test was used. Data without normal distribution were analysed with the nonparametric U Mann– Whitney method. The results were considered to be statistically significantly different at a probability level of *p* < 0.05.

## 5. Conclusions

Based on the results of the present study, it could be concluded that EXTs and PhAs are safe at tested concentrations. The cytotoxicity assays showed low toxicity towards V79 cells, and the *umu*-test indicated no genotoxic or photo-genotoxic activity and even a photo-protective effect. Moreover, tested EXTs were effective towards 2AA-induced genotoxicity and exhibited low antigenotoxic effect against MMC. The current investigations pointed out that the observed EXTs properties could not be attributed only to the one group of compounds present in tested EXTs, as tested PhAs proved to be inactive or slightly active in the performed *umu*-test but their activity was based on combined effectiveness of all EXTs’ constituents. The results of the present study indicates the safety and photo-protective potential of plant extracts derived from *P. filicifolia* shoots cultivated in vitro and could serve as a basis for their future application as a source of plant stem cells in cosmetic formulation. Although, the tested extracts showed promising antigenotoxic and anti-photogenotoxic potential, further studies should be conducted to assess their activity and safety towards mammalian cells in vitro and in vivo.

## Figures and Tables

**Figure 1 molecules-25-01090-f001:**
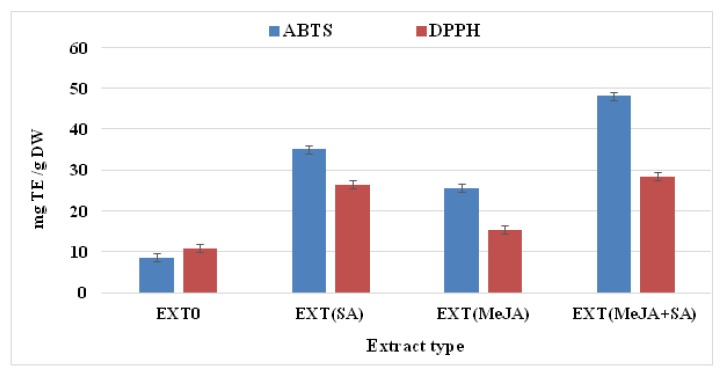
Antioxidant properties assessed using ABTS and DPPH methods for extracts obtained from shoots of *P. filicifolia* cultivated under various elicitor treatments: salicylic acid (SA, 50 μM), methyl jasmonate (MeJA, 200 μM), and (MeJA 200 μM + SA 50 μM): Data are expressed as means ± SD from two independent experiments performed in triplicate. All differences among various extract types in antioxidant activity are statistically significant at *p*< 0.05.

**Figure 2 molecules-25-01090-f002:**
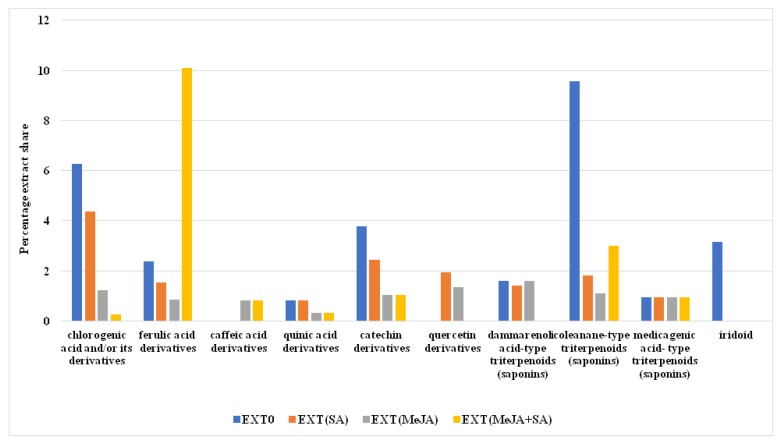
The percentage share of groups of chemical constituents in tested methanolic extracts of *P. filicifolia* determined on the basis of the UHPLC-DAD-MS/MS analysis.

**Figure 3 molecules-25-01090-f003:**
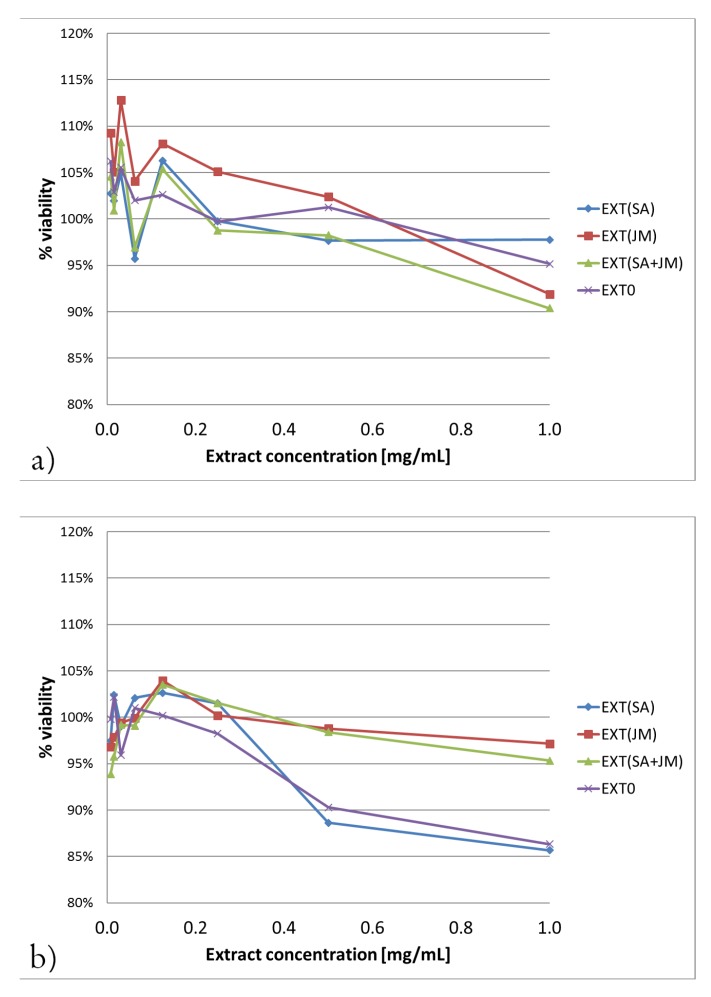
Results of two different cytotoxicity assays: (**a**) Neutral Red Uptake (NRU) (**b**) MTT.

**Table 1 molecules-25-01090-t001:** The content (mg/g DW) of phenolic acids: chlorogenic (CGA), caffeic (CA), ferulic acid (FA), total phenolics (GAE mg/g DW), and total flavonoids (QE mg/g DW) in elicited shoots of *P. filcifolia* cultivated in vitro.

Extract Type	CGA mg/g DW	CA mg/g DW	FA mg/g DW	Total Content Phenolics mg GAE/g DW	Total Content Flavonoids mg QE/g DW
EXT0	1.18 ± 0.07 *	0.6 ± 0.13	0.02± 0.001	1.5 ± 0.35 *	1.0 ± 0.17 *
EXT(SA)	2.8 ± 0.13	0.8 ± 0.22	0.07 ± 0.08	3.9 ± 0.53 *	2.5 ± 0.35 *
EXT(MeJA)	3.1 ± 0.24	0. 8 ± 0.29	0.09 ± 0.01	5.3 ± 0.49 *	3.4 ± 0.35 *
EXT(MeJA+SA)	4.2± 0.17 *	0.9 ± 0.23	0.1 ± 0.23	7.8 ± 0.21 *	4.3 ± 0.27 *

EXT0—extract of control shoots cultivated on LS medium without elicitors; EXT(SA)—extracts of shoots elicited with salicylic acid (50 μM); EXT(MeJA)—extracts of shoots elicited with methyl jasmonate (200 μM); EXT(MeJA + SA)—extracts of shoots elicited with salicylic acid and methyl jasmonate (200 and 50 μM, respectively). Data are expressed as means ± SD from two independent experiments performed in triplicate. Asterisk (*) indicate significant differences at *p* < 0.05 among various extract types.

**Table 2 molecules-25-01090-t002:** Evaluation of genotoxic activity of the tested extracts based on IR values.

IR (mean ± SD)	Tested Material Concentration [mg/mL]				
	**0.125**	**0.25**	**0.5**	**1**	2
EXT0 (−S9) EXT0 (+S9) EXT(SA) (−S9) EXT(SA) (+S9) EXT(MeJA) (−S9) EXT(MeJA) (+S9) EXT(SA+MeJA) (−S9) EXT(SA+MeJA) (+S9)	0.9 ± 0.10	1.0 ± 0.25 1.1 ± 0.20 0.96 ± 0.08 0.99 ± 0.07 0.7 ± 0.89 0.97 ± 0.08 0.7 ± 0.77 1.0 ± 0.13	1.1 ± 0.43 1.05 ± 0.03 1.17 ± 0.07 1.04 ± 0.05 1.4 ± 0.28 0.99 ± 0.06 1.1 ± 0.16 1.05 ± 0.05	1.0 ± 0.16 1.05 ± 0.06 1.16 ± 0.05 1.1 ± 0.15 1.3 ± 0.19 1.01 ± 0.08 1.2 ± 0.20 1.00 ± 0.02	0.97 ± 0.08 1.18 ± 0.09 1.2 ± 0.31 1.16 ± 0.09 1.0 ± 0.16 1.20 ± 0.06 1.2 ± 0.25

Each value is expressed as mean ± standard deviation from at least 3 independent biological evaluation.

**Table 3 molecules-25-01090-t003:** Evaluation of genotoxic activity of the phenolic acids based on IR values of the highest tested concentration (10 μg/mL).

	IR (mean±SD)
CGA (−S9) CGA (+S9) FA (−S9) FA (+S9) CA (−S9) CA (+S9)	1.3 ± 0.19 1.0 ± 0.21 1.2 ± 0.10 1.1 ± 0.14 1.1 ± 0.16 1.0 ± 0.11

Each value is expressed as mean ± standard deviation from at least 3 independent biological evaluations.

**Table 4 molecules-25-01090-t004:** Evaluation of antigenotoxic activity based on IR values for the control—genotoxins 4-nitroquinoline-N-oxide (4NQO) and mitomycin C (MMC)—and the mixture of genotoxins with the tested compounds.

Concentration of EXTs [mg/mL], PhAs [μg/mL]	4NQO 0.25 [μg/mL]		4NQO 0.05 [μg/mL]		MMC 0.02 [μg/mL]		MMC 0.004 [μg/mL]	
	IR (mean ± SD)	Antigenotox.	IR (mean ± SD)	Antigenotox.	IR (mean ± SD)	Antigenotox.	IR (mean ± SD)	Antigenotox.
0 (control) EXT0 2 1 0.5 0.25 EXT(SA) 2 1 0.5 0.25 0.125 EXT(MeJA) 2 1 0.5 0.25 EXT(MeJA+SA) 2 1 0.5 0.25 CGA 10 5 2.5 1.25 0.625 0.3125 FA 10 5 2.5 1.25 0.625 0.3125 CA 10 5 2.5 1.25 0.625 0.3125	9 ± 2.6 8 ± 1.2 10 ± 2.7 8 ± 2.7 7 ± 1.4* not tested 11 ± 3.0* 10 ± 2.5 7 ± 1.7* 8 ± 1.6 13 ± 2.4* 11 ± 2.6* 12 ± 2.2* 14 ± 4.1* 13 ± 1.2* 11 ± 2.9* 11 ± 5.1 11 ± 5.1 11 ± 2.6 11 ± 3.0 10 ± 1.4 9 ± 1.8 10 ± 3.0 8 ± 2.8 12 ± 2.2 11 ± 2.5 9 ± 1.5 9 ± 1.7 8 ± 2.4 8 ± 2.5 11 ± 1.8 10 ± 1.5 9 ± 1.2 8 ± 1.6 8 ± 2.3 7 ± 2.1*	- 20% −25% 20% −47% −26% −32% −53% −45% −24% 25%	3.7 ± 0.99 3.7 ± 0.52 4.3 ± 0.96* 3.5 ± 0.83 3.0 ± 0.36 not tested 5.2 ± 0.56* 4.3 ± 0.66 3.1 ± 0.34 2.9 ± 0.51 4.9 ± 0.42* 4 ± 1.0 4 ± 1.2 4 ± 1.4 4.8 ± 0.46* 4.9 ± 0.31* 5.1 ± 0.33* 5.4 ± 0.37* 4.1 ± 0.96 3.9 ± 0.83 4.0 ± 0.85 3.7 ± 0.78 3.4 ± 0.79 3.8 ± 0.96 4.0 ± 0.70 3.6 ± 0.67 3.6 ± 0.75 3.8 ± 0.75 3.5 ± 0.77 3.9 ± 0.97 3.9 ± 0.99 3.7 ± 0.69 3.6 ± 0.47 3.6 ± 0.78 3.3 ± 0.64 4 ± 1.0	- −17% −43% −33% −30% −35% −39% −48%	5.3 ± 0.70 6 ± 1.0 6 ± 1.5 6 ± 1.3 6 ± 1.3 5 ± 1.2 4.9 ± 0.94* 5 ± 1.1* 6 ± 1.2 not tested 5 ± 1.6* 5 ± 1.4 6 ± 1.1 7 ± 1.5* 5 ± 1.7 5 ± 1.0 6 ± 1.5 6 ± 1.0 4 ± 1.1 4 ± 1.5 5 ± 1.1 4.5 ± 0.86 5 ± 1.1 5 ± 1.1 5 ± 1.2 5 ± 1.2 5 ± 1.2 4.8 ± 0.71 5 ± 1.1* 5.2 ± 0.55 5 ± 1.2 5 ± 1.1 5 ± 1.2 5 ± 1.1 5.1 ± 0.91 5.0 ± 0.44	- 16% 16% 14% −16% −14%	2.4 ± 0.42 2.1 ± 0.39 2.4 ± 0.47 2.3 ± 0.32 2.0 ± 0.13* 2.3 ± 0.27 2.4 ± 0.39 2.2 ± 0.40 2.4 ± 0.45 not tested 2.2 ± 0.40 2.3 ± 0.37 2.5 ± 0.43 2.6 ± 0.41 2.5 ± 0.53 2.6 ± 0.34 2.6 ± 0.38 2.6 ± 0.28 2.2 ± 0.40 2.4 ± 0.55 2.5 ± 0.71 2.7 ± 0.85 2.6 ± 0.81 2.2 ± 0.57 2.3 ± 0.51 2.5 ± 0.62 2.6 ± 0.93 2.4 ± 0.71 2.5 ± 0.98 2.2 ± 0.70 2.3 ± 0.67 2.7 ± 0.81 2.5 ± 0.75 2.4 ± 0.88 2.6 ± 0.80 2.6 ± 0.74	- 17%

IR values of the control are expressed as mean ± standard deviation of all experiments. Asterisk (*) indicate significant differences at *p* < 0.05. A significant difference is calculated between IR values of the control measured in the same experiment as tested mixtures. Each value is expressed as mean ± standard deviation from at least 3 independent biological evaluations.

**Table 5 molecules-25-01090-t005:** Evaluation of antigenotoxic activity based on IR values for the control—genotoxins 2-aminoanthracene (2AA)—and the mixture of genotoxins with the tested extracts.

Tested Extracts Concentration (mg/mL)	2AA 10 (μg/mL)		2AA 1 (μg/mL)	
	IR (mean ± SD)	Antigenotox.	IR (mean ± SD)	Antigenotox.
0 (control) EXT0 2 1 0.5 0.25 EXT(SA) 2 1 0.5 0.25 EXT(MeJA) 2 1 0.5 0.25 EXT(MeJA+SA) 2 1 0.5 0.25	6 ± 1.5 3.6 ± 0.67* 5.8 ± 0.86 6 ± 1.4 6 ± 1.6 3 ± 1.1* 4 ± 1.2* 5 ± 1.5* 5 ± 1.1 2.8 ± 0.42* 3.7 ± 0.70* 4.9 ± 0.87* 5 ± 1.2 2.0 ± 0.24* 3.2 ± 0.34* 4.3 ± 0.45* 4.7 ± 0.95*	- 38% 43% 29% 16% 51% 36% 16% 66% 45% 26% 19%	4.5 ± 0.99 2.9 ± 0.77* 4.4 ± 0.57 4 ± 1.0 5 ± 1.4 2.6 ± 0.89* 3.4 ± 0.88* 4.1 ± 0.93 4.4 ± 0.56 2.4 ± 0.21* 3.2 ± 0.56* 3.8 ± 0.75* 4 ± 1.2 1.7 ± 0.27* 2.7 ± 0.34* 3.4 ± 0.46* 4.1 ± 0.57	- 35% 41% 24% 45% 28% 16% 63% 39% 24%

**Table 6 molecules-25-01090-t006:** Evaluation of antigenotoxic activity based on IR values for the control—genotoxins 2AA—and the mixture of genotoxins with PhAs.

Tested Extracts Concentration (mg/mL)	2AA 10 (μg/mL)		2AA 1 (μg/mL)	
	IR (mean ± SD)	Antigenotox.	IR (mean ± SD)	Antigenotox.
0 (control) CGA 10 5 2.5 1.25 0.625 0.3125 FA 10 5 2.5 1.25 0.625 0.3125 CA 10 5 2.5 1.25 0.625 0.3125	4 ± 1.3 4.6 ± 0.66* 5 ± 1.5* 3.3 ± 0.80 3.2 ± 0.77 3.2 ± 0.88 2.9 ± 0.43 5 ± 1.4 4 ± 1.3 3.2 ± 0.45 3.3 ± 0.63 3.1 ± 0.75 2.9 ± 0.43 3.7 ± 0.80 3.6 ± 0.89 3.2 ± 0.51 2.7 ± 0.34* 3.4 ± 0.90 2.8 ± 0.60	- −29% −36% 24%	3.37 ± 0.72 4 ± 1.2* 5 ± 1.4* 3.2 ± 0.47 3.6 ± 0.95 3.1 ± 0.60 3 ± 1.0 4 ± 1.0 3.4 ± 0.96 2.8 ± 0.58 2.9 ± 0.66 2.7 ± 0.37* 3.1 ± 0.73 3.8 ± 0.85 3.7 ± 0.77 3.1 ± 0.53 3.1 ± 0.29 3.1 ± 0.55 3 ± 1.0	- −27% −34% 19%

Each value is expressed as mean ± standard deviation from at least 3 independent biological evaluations. Asterisk (*) indicate significant differences at *p* < 0.05.

**Table 7 molecules-25-01090-t007:** Evaluation of anti-photogenotoxic activity based on IR values for the control—photogenotoxin chlorpromazine (CPZ)—and the mixture of CPZ with EXT(MeJA + SA) and PhAs.

Tested Concentration EXT (mg/mL) PhAs (μg/mL)	CPZ 10 (μg/mL)		CPZ 5 (μg/mL)	
	IR (mean ± SD)	Anti-Photogenotox.	IR (mean ± SD)	Anti-Photogenotox.
0 (control) EXT(MeJA + SA) 1 0.5 0.25 0.125 CGA 10 5 2.5 1.25 FA 10 5 2.5 1.25 CA 10 5 2.5 1.25	5.1 ± 0.78 3.3 ± 0.65* 4.6 ± 0.75 5 ± 1.0 4 ± 1.3* 4.8 ± 0.43 5.0 ± 0.70 4.5 ± 0.58* 4.6 ± 0.83 4.3 ± 0.46* 4.5 ± 0.73 4 ± 1.1* 4.1 ± 0.97* 4.9 ± 0.90 5 ± 1.2 4.5 ± 0.49* 4.9 ± 0.62	- 34% 14% 11% 15% 14% 18% 10%	3.6 ± 0.40 2.3 ± 0.38* 3.1 ± 0.58* 3.0 ± 0.80* 3.2 ± 0.82 3.4 ± 0.26 3.3 ± 0.34 3.1 ± 0.53* 3.3 ± 0.50 3.2 ± 0.20* 3.0 ± 0.37* 3.0 ± 0.41* 3.0 ± 0.43* 3.4 ± 0.25 3.3 ± 0.33 3.2 ± 0.40* 3.4 ± 0.54	- 37% 13% 16% 13% 11% 15% 20% 19% 11%

Each value is expressed as mean ± standard deviation from at least 3 independent biological evaluations. Asterisk (*) indicate significant differences at *p* < 0.05.

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
