# Peer review of "Antigenotoxic, Anti-photogenotoxic, and Antioxidant Properties of Polyscias filicifolia Shoots Cultivated In Vitro"

_molecules, 2020, doi:10.3390/molecules25051090_

Round 1

Reviewer 1 Report

The manuscript is quite preliminary ... the absence of in vivo testing does not allow confirming the authors' claims. Cytotoxic and antioxidant activities are common in plants and of little relevance in vitro.

Author Response

Dear Reviewer,

Firstly, we would like to thank the reviewer for careful and thorough reading of this manuscript and for the thoughtful comments and constructive suggestions, which help to improve the quality of this manuscript. We have modified the text according to the suggestions made and provide below responses to the comments:

Comment: The manuscript is quite preliminary ... the absence of in vivo testing does not allow confirming the authors' claims. Cytotoxic and antioxidant activities are common in plants and of little relevance in vitro.

Reply: We agree that the presented results are just the first step in the process of the evaluation of the antigenotoxic potential of the tested extracts. The next step should be to conduct in vitro and in vivo assays. Moreover, further research should be conducted to evaluate the mechanism of antigenotoxic action. In the conclusions section we added the information about the necessity of further research.

We very much hope the revised manuscript is accepted for publication in Journal.

Sincerely yours,

Ramona Figat.

Reviewer 2 Report

1. The term "DW" is not generally accepted, as far as I know, so I recommend to decipher it.

2. Adaptogens are very dubious group of drugs, you reduce greatly the scientific value of your article mentioning them

Author Response

Dear Reviewer,

Firstly, we would like to thank the reviewer for careful and thorough reading of this manuscript and for the thoughtful comments and constructive suggestions, which help to improve the quality of this manuscript. We have modified the text according to the suggestions made and provide below responses to the comments:

Comment: The term "DW" is not generally accepted, as far as I know, so I recommend to decipher it.

Reply: We deciphered the abbreviation (line 83)

Comment: Adaptogens are very dubious group of drugs, you reduce greatly the scientific value of your article mentioning them.

Reply: We mentioned adaptogenic activity of Polyscias filicifolia according to the cited literature.

We very much hope the revised manuscript is accepted for publication in Journal.

Sincerely yours,

Ramona Figat.

Reviewer 3 Report

The manuscript is interesting, using the good technology. There are some errors in the use of significant figures, which should be thoroughly reviewed.

Suggestions to correct are highlighted in the yellow text. It is recommended to homogenize the writing style.

An example chromatogam and the UV and MS spectra of the identified major compounds can be provided in the support material.

Author Response

Dear Reviewer,

Firstly, we would like to thank the reviewer for careful and thorough reading of this manuscript and for the thoughtful comments and constructive suggestions, which help to improve the quality of this manuscript. We have modified the text according to the suggestions made and provide below responses to the comments:

Comment: There are some errors in the use of significant figures, which should be thoroughly reviewed.

Reply: We corrected significative figures in the whole manuscript

Comment: Suggestions to correct are highlighted in the yellow text. It is recommended to homogenize the writing style.

Reply: We checked the highlighted suggestions and we corrected the text.

Comment: Standard deviation is bigger than the data.

Reply: In the present experiments we used the bacterial assay. The relatively large standard deviation (in data presented in Table 2) could be explained by differences in bacterial response. The umu-test procedure have a lot of steps  important for β-galactosidase activity. The measured β-galactosidase activity is a sensitive factor. IR was calculated on the basis of spectrophotometric measurement. The absorption at 420 nm indicated the intensity of enzymatic reaction. The 1.5 fold and greater increase of the β-galactosidase activity resulting in IR  value of 1.5 and greater indicated the genotoxicity of the sample. In Table 2 we have IR value lower than 1.5, which indicates that the β-galactosidase activity was very low.

Sincerely yours,

Ramona Figat.

Round 2

Reviewer 1 Report

As said before, the manuscript is quite preliminary. The absence of in vivo testing does not allow confirming the authors' claims. Cytotoxic and antioxidant activities are usual in plants and of little relevance in vitro. The authors do not make any real change in the text, so I maintain my position by rejecting the publication.